# Stretchable Woven Fabric-Based Triboelectric Nanogenerator for Energy Harvesting and Self-Powered Sensing

**DOI:** 10.3390/nano13050863

**Published:** 2023-02-25

**Authors:** Lijun Chen, Tairan Wang, Yunchu Shen, Fumei Wang, Chaoyu Chen

**Affiliations:** 1Engineering Research Center of Knitting Technology, Ministry of Education, College of Textile Science and Engineering, Jiangnan University, Wuxi 214122, China; 2Key Laboratory of Textile Science & Technology, Ministry of Education, College of Textiles, Donghua University, Shanghai 201620, China

**Keywords:** woven fabric, stretchable sensing, energy harvesting, triboelectric nanogenerators

## Abstract

With the triboelectric nanogenerator developing in recent years, it has gradually become a promising alternative to fossil energy and batteries. Its rapid advancements also promote the combination of triboelectric nanogenerators and textiles. However, the limited stretchability of fabric-based triboelectric nanogenerators hindered their development in wearable electronic devices. Here, in combination with the polyamide (PA) conductive yarn, polyester multifilament, and polyurethane yarn, a highly stretchable woven fabric-based triboelectric nanogenerator (SWF-TENG) with the three elementary weaves is developed. Different from the normal woven fabric without elasticity, the loom tension of the elastic warp yarn is much larger than non-elastic warp yarn in the weaving process, which results in the high elasticity of the woven fabric coming from the loom. Based on the unique and creative woven method, SWF-TENGs are qualified with excellent stretchability (up to 300%), flexibility, comfortability, and excellent mechanical stability. It also exhibits good sensitivity and fast responsibility to the external tensile strain, which can be used as a bend–stretch sensor to detect and identify human gait. Its collected power under pressure mode is capable of lighting up 34 light-emitting diodes (LEDs) by only hand-tapping the fabric. SWF-TENG can be mass-manufactured by using the weaving machine, which decreases fabricating costs and accelerates industrialization. Based on these merits, this work provides a promising direction toward stretchable fabric-based TENGs with wide applications in wearable electronics, including energy harvesting and self-powered sensing.

## 1. Introduction

Environmental damage threatens the whole of civilization. The world needs alternative energy sources urgently, such as solar energy, water, geothermal energy, wind energy, marine energy, and biomass energy. Based on a coupled effect of contact-electrification and electrostatic induction, a triboelectric nanogenerator (TENG) [1,2,3,4,5] is a newly developed energy-harvesting technology. TENGs use organic materials so that they are light, smaller, and cost-effective. The output characteristics of TENGs are high voltage but low current [6,7]. Due to the output characteristics, TENG can directly convert a mechanical triggering into a self-generated electric signal for the detection of motion, vibration, mechanical stimuli, physical touching, and biological movement [5]. It is not only capable of harvesting dispersed energy that has been wasted these years, such as human motion [8,9,10], vibration energy (car tire) [11,12,13], and water waves [14,15,16], but also can be applied in the self-powered sensors field, such as the smart textiles [17,18,19] and artificial intelligent electronics [20]. With all these merits, TENGs attract the attention of researchers from different fields, especially the textile material field. Various textile-based TENGs, including fiber-based TENG, yarn-based TENG, and fabric-based TENG, have been applied in physiological monitors [21], artificial sensors [22], human–interactive interfaces [23], smart clothes [24], and so on.

With the rapid development of wearable electronics, the requirements of stretchability [25,26], low cost, high efficiency, lightweight, comfortability, and flexibility are receiving intensive attention. Nevertheless, most textiles cannot meet the stretchable requirement and the mass-manufacturing technology requirement which largely hindered their development. Although significant progress has been made in improving the stretchability of textile-based TENGs [27,28,29,30], the limited manufacturing method without innovation still faces several critical challenges. Firstly, due to its high stretchability and shape-adaption the knitting structure is applied for fabricating the stretchable fabric-based TENG [8]. However, some knitting fabrics have low elastic resilience, which affects the service life. Integrating different elastomeric polymer materials layer by layer to fabricate the stretchable fabric-based TENG is the other method that sacrifices the thickness, size, the comfortability, and has to use the complex manufacturing process [31]. Secondly, some researchers achieve stretchability by sewing the wave-shaped conductive fabric or serpentine shape conductive yarn on an elastic fabric [24,32,33], which is difficult for industrial mass production. Thirdly, winding the conductive yarn densely around the elastic yarn is the main method to make the stretchable core-sheath fiber-based and yarn-based TENG. The core-sheath structure [34,35] is comprised of several different materials outside the elastic core yarn, which not only increase the yarn diameter and the cost but also decrease the structure’s stability and comfortability. Therefore, a new manufacturing method needs to be developed and simplified to solve these challenges.

In this work, in combination with the polyamide (PA) conductive yarn, polyester (PET) multifilament, and polyurethane (PU) yarn, a highly stretchable woven fabric-based triboelectric nanogenerator (SWF-TENG) with the three elementary weaves (including plain, twill, and satin weave) to be designed and presented, is capable of harvesting mechanical energy and tensile strain sensing. TENG has four basic working modes, including the vertical contact-separation mode, the contact-sliding mode, the single electrode mode, and the freestanding triboelectric-layer mode. The single electrode mode is applied in the SWF-TENG. The PA conductive yarn is chosen as the electrode, and the other electrode (the ground) is just a reference electrode as a source for electrons. Based on the mature textile manufacturing technology, woven fabric is easy to fabricate with high efficiency. However, it is hard for the woven fabric to achieve high stretchability due to the natural interior mutual interlacing yarns. Therefore, the new design improvements based on the woven structure have been made, including two key points. One is controlling the weft yarn density before and after coming from the loom. The other one is choosing elastic yarn as a part of warp yarns. PU yarn, PA conductive yarn, and PET multifilament are chosen as the warp yarns. PET multifilament is chosen as the weft yarns. Due to the most common yarns applied and the unique woven structure, the SWF-TENG is cost-effective, flexible, stretchable, comfortable, and can respond rapidly to external mechanical stimuli that can be used for self-powered sensing and energy harvesting. In addition, the SWF-TENG has a high contacting-separating area between yarns during stretching–releasing motion which does not need to rely on external substrates or surfaces to acquire sliding or contact-separating triboelectricity. It can generate electricity by only stretching itself. As an energy harvester, the SWF-TENG is capable of lighting up 34 LEDs marked as alphabet “KTC” by only hand tapping the fabric. As a bend–stretch sensor, the SWF-TENG can detect and identify human gait. Our work not only presents a creative, mass-manufactured, and highly stretchable fabric-based TENG, which has stable outputs during stretching and releasing motion, but also provides a promising new direction and design mentality toward fabricating the stretchable textile-based TENGs.

## 2. Materials and Methods

Materials: Based on the most common weaving technology, several SWF-TENGs with different fabric weaves are fabricated, including plain weave, twill weave, and satin weave. In these SWF-TENGs, a commercially purchased 12tex/72F PET multifilament is chosen as the dielectric material of the SWF-TENG due to its low-cost, fluffy structure, and a strong tendency to lose electrons. A two-ply polyamide (PA) yarn coated with Ag is used as the electrode of the SWF-TENG, owing to its excellent conductivity and good softness. The yarn fineness is 280D. The super elastic polyurethane (PU) yarn with a width of 0.3 mm is chosen as the warp yarn of all woven fabrics.

Fabrication of the woven fabric: The SWF-TENG is easily fabricated into a plain woven pattern with a miniature loom by interlacing warp yarns (PET/PU/PA conductive yarn or PET/PU) with weft yarns (PA conductive yarn or PET multifilament) fabricated by a weaving machine. For the common woven fabric the warp yarns must be parallel and under uniform tension; it is the purpose of warping. However, for the SWF-TENG the PU elastic yarns as part of the warp yarns are stretched to 300% during the weaving process. The other warp yarns without stretchability are just kept straight by the weaver’s beam and take-up cloth roller together. Therefore, the fabric will retract to the normal length after coming from the loom. The weft yarn density during the weaving process is adjusted to a relatively low value, 83 yarns per 10 cm. Due to the low weft yarn density, the distance between the adjacent weft yarn is big enough. Based on this unique manufacturing method and the high elastic resilience of PU yarn, the fabric will be shrunk along the warp direction while the PU recovers to its original state after finishing weaving. The shrinkage of the whole fabric leads to the buckling of other warp yarns floating as loops on the fabric surface, including the PET multifilament and PA conductive yarn. Therefore, the weft yarn density after coming from the loom is 250 yarns per 10 cm.

The surface morphologies of the silver-coated PA yarns and PET multifilaments are characterized by a field emission scanning electron microscope (SEM) and an optical microscope (OM). The mechanical tensile test was conducted by a universal mechanical testing machine (Model, Instron 5567). The electrical output performances (V_OC_, I_SC_, and Q_SC_) of the TENG during compressing and stretching operations were implemented by a linear motor and an electrometer (Keithley 6514 System, Tektronix, America). The compressing forces were measured by Vernier LabQuest Mini.

## 3. Results and Discussion

The scanning electron microscopy (SEM) photograph of the PA conductive yarn is presented in Figure 1a. Figure 1b presents the partially enlarged view of the yarn’s surface. Through Figure 1b it can be found that the Ag coating is uniform on the surface of each fiber in the PA yarn, not the surface of the yarn. All fibers with Ag coating are twisted into the PA conductive yarn, which endows the yarn with flexibility and conductivity. Its tensile property is shown in Figure 1c, with the elongation at break of 33.69% and a breaking strength of 3.25 cN/dtex. Figure 1d presents the photograph of the PET multifilament which is flexible and can stand different mechanical deformations. The OM images of the PET multifilament at the original state without stretch (upper view) and the stretch state 12% tensile strain (lower view) are shown in Figure 1e. This comparison demonstrates the great fluffiness performance of the PET multifilament, which could improve the contact separation area with the PA conductive yarn (electrode). To show the fluffy state clearly, the SEM image is also presented (Figure 1f).

Due to the unique fabricating method and the large tensile strain of PU during the weaving process, the PU warp interlacing points are invisible on the surface of SWF-TENG. As shown in Figure 2a–c, the digital photographs of the satin weave are presented. The large tensile strain of PU leads to one gap (Figure 2b,d) existing between adjacent weft weave repeat units. The PET and PA conductive yarn beside the gap will contact and separate from each other during the stretching–releasing motion. The back side of SWF-TENG with satin weave is covered with loops of PET filaments and PA conductive yarn (Figure 2e). Through the weaving state (Figure 2c), the SWF-TENG has lower weft yarn density which resulted in the large elasticity of the fabric. According to the weave diagram of SWF-TENG with satin weave (Figure 2f), the shift in the warp direction is two and one warp weave repeat unit is five. In one warp weave repeat unit, the yarn arrangement is PU/PA/PET/PA/PET.

PET and PA are contacting before stretch and separating after stretch, which is illustrated in Figure 2g–i. To make it clear, the digital photograph of the SWF-TENG with satin weave in a stretch state is presented (Figure 2h,i). In a stretched state (Figure 2h,i), the warp interlacing points, including PU/PA/PA/PET/PET, are exposed in the surface of the first/second/third/fourth/fifth line in one repeat unit. In an un-stretched state (Figure 2g), not only the PU warp interlacing points (in the first line of one weft weave repeat unit), but also the PA conductive yarn warp interlacing points (in the second line of one weft weave repeat unit) and the PET (in the fifth line of one weft weave repeat unit) are hidden in the gap and invisible on the fabric surface.

The working mechanism of SWF-TENG with satin weave under stretching mode is schematically illustrated in Figure 3a. In the original stage (Figure 3(ai)), the second line and the third line in one weft weave repeat unit contact with the fourth line and fifth line in the adjacent weft weave repeat unit. The surfaces of the PA conductive yarn warp interlacing points and PET warp interlacing points contact with each other and are charged with the same number of opposite charges. The PA conductive yarn proves to be negatively charged because of the Ag coating’s ability to attract electrons. There is practically no electrical potential difference between the two surfaces due to the two opposite charges coinciding at the same plane. By stretching the SWF-TENG gradually (Figure 3(aii)), the PU warp interlacing points (in the first line of one weft weave repeat unit), the PA conductive warp interlacing points (in the second line of one weft weave repeat unit), and the PET (in the fifth line of one weft weave repeat unit) which are hidden in the gap will gradually emerge on the fabric surface. Therefore, the PA conductive yarn and PET get separated gradually, leading to a change in the electrode (PA conductive yarn) potential. A potential difference between the electrode and the ground prompts electrons to flow from the electrode through external loading to the ground, resulting in an electrical current. As these two surfaces are moving quite far away a new electrical equilibrium achieves, and the electrons stop moving (Figure 3(aiii)). On removal of the tensile strain, both the PA conductive yarn and PET recover to the initial starting configuration gradually. The PA conductive yarn approaches PET again, electrons flow inversely from the electrode (PA conductive yarn) to the ground to achieve a charge balance (Figure 3(aiv)). When the PA conductive yarn contacts the PET, charge neutralization occurs again. Continuous contact-separation movements between the PA conductive yarn and the PET generate continuous alternating current outputs from the SWF-TENG through the external loading. Figure 3b,c presents the V_OC_ and I_SC_ of the SWF-TENG at different tensile strains, ranging from 20% to 100%.

Different from the SWF-TENG with a satin weave, the working mechanism of the twill weave structure has changed. As shown in Figure 4a–e, the digital photographs of the SWF-TENG with twill weave are presented. Same as with a satin weave, the twill weave structure also has a gap that is visible at the stretching state and invisible at the original state (Figure 4b,d). It exists between two adjacent weft weave repeat units. The PET and PA conductive yarn beside the gap will contact and separate from each other during stretching–releasing motion, which generated electricity. The back side of SWF-TENG with twill weave is covered with loops of PET filaments and PA conductive yarn (Figure 4e). The weave diagram of SWF-TENG with twill weave is shown in Figure 1f, in which a warp weave repeat unit is five. In one warp weave repeat unit, the yarn arrangement is PU/PA/PET/PA/PET. However, the contact area is different between the satin weave structure and the twill weave structure. As shown in Figure 4a–c, the PET (in the fifth line of one weft weave repeat unit) and the PA conductive yarn warp interlacing points (in the second line of one weft weave repeat unit) are invisible before stretch, but visible after stretch. Therefore, only half of PA conductive yarn warp interlacing points (in the second line of one weft weave repeat unit) in one weft weave repeat unit can contact and separate with PET yarn in the adjacent weft weave repeat unit.

The working mechanism is presented in Figure 5a. In the original stage (Figure 5(ai)), the second line (warp interlacing points of PA conductive yarn) in one weft weave repeat unit contacts with the fourth line (warp interlacing points of PA conductive yarn) and the fifth line (warp interlacing points of PE conductive yarn) in the adjacent weft weave repeat unit. The surfaces of the PA conductive yarn warp interlacing points and PET warp interlacing points contact with each other and are charged with the same number of opposite charges. There is practically no electrical potential difference between the two surfaces due to the two opposite charges coinciding at the same plane. By stretching the SWF-TENG gradually (Figure 5(aii)), the PU warp interlacing points (in the fifth line of one weft weave repeat unit) which are hidden in the gap will gradually emerge on the fabric surface. Therefore, the PA conductive yarn and PET get separated gradually, leading to a change in the electrode (PA conductive yarn) potential. The potential difference between the electrode and the ground prompts electrons to flow from the electrode through external loading to the ground, resulting in an electrical current. As these two surfaces are moving quite far away, a new electrical equilibrium achieves and the electrons stop moving (Figure 5(aiii)). On removal of the tensile strain, both the PA conductive yarn and PET recover to the initial starting configuration gradually. The PA conductive yarn approaches PET again, and electrons flow inversely from the electrode (PA conductive yarn) to the ground to achieve a charge balance (Figure 5(aiv)). When the PA conductive yarn contacts the PET charge neutralization occurs again. Figure 5b,c presents the V_OC_ and I_SC_ of the SWF-TENG at different tensile strains, ranging from 20% to 100%.

Besides the SWF-TENGs with satin weave and the twill weave, the SWF-TENG with a plain structure is also fabricated. Figure 6a,b presents photographs of the SWF-TENG with a plain structure before and after the stretch. Through the pictures no gap exists between two adjacent weft weave repeat units, which is different from those two woven structures (satin weave and twill weave). The weave diagram of the SWF-TENG is shown in Figure 6c, in which one weft weave repeat unit is 2. The bucking part of PA conductive yarn and PET yarn along the warp direction contact and separate from each other during stretching mode. The working mechanism is presented in Figure 6d. The PA conductive yarn warp interlacing points in one weave repeat unit will contact and separate with the PET filaments warp interlacing points in the adjacent weave repeat unit during the stretching–releasing process, which generates continuous alternating current outputs through the external loading. Figure 6e,f presents the V_OC_ and I_SC_ of the SWF-TENG at different tensile strains, ranging from 20% to 100%. To characterize the tensile strain property of each fabric the tensile test is designed, which is shown in Figure 6g. The distance between the two clamp holders is adjusted to 5 cm. The elastic modulus of each SWF-TENG with a satin weave, twill weave, and plain weave is 6.868 N/cm, 7.506 N/cm, and 9.478 N/cm. The elastic moduli of the three weave structures are low at a tensile strain of 5%. The fabric with a satin weave has the lowest elastic modulus.

According to the output results of SWF-TENGs with satin, twill, and plain structures under various tensile strains, the electrical output performances gradually increase with the increase in tensile strain, demonstrating the high stretch sensitivity of the SWF-TENG. The stable electrical output performance and high stretch sensitivity of the yarn-based TENG are attributed to the elastic performance and the stable woven structure. According to the comparison of these three SWF-TENGs’ outputs, which is shown in Figure 7a, it is obvious that SWF-TENG with satin weave has the highest electrical output performance and SWF-TENG with plain weave has the lowest electrical output performance. Among them, the gap existing between the adjacent weft weave repeat units plays a key role, which affects the contact area between the PA conductive yarn and PET.

To study the sensor’s sensitivity of the SWF-TENG in-depth, its electrical output voltage (Figure 7a) as a function of the tensile strains is investigated. The R^2^ of the linear fitting equation in each structure (satin, twill, and weave) outputs are 0.8686, 0.99532, and 0.91755, respectively. The V_OC_ signal profiles of the SWF-TENG with twill weave are regular when the tensile strain is below 100%. Although the SWF-TENG with satin weave has higher outputs, the SWF-TENG with twill weave has higher accuracy to the tensile strain. For the SWF-TENG with twill weave, the slope of the linear fit curve is 0.0618. Therefore, the sensitivity of the SWF-TENG is 0.0618 V/cm. It is more suitable to be used as a stretchable sensor, which can predict the tensile strain through the outputs.

SWF-TENGs with different weave structures have different contact areas. As indicated above, for the satin weave, the PET warp interlacing points (the fourth line and fifth line of one weft weave repeat unit) in one adjacent weft weave repeat unit and the PA conductive yarn warp interlacing points (the second line and the third line of one weft weave repeat unit) in the adjacent weft weave repeat unit are contacting and separating with each other during stretch–release motion. For twill weave, the PA conductive yarn warp interlacing points (the second line of one weft weave repeat unit) in one weft weave repeat unit contact and separate with the PET warp interlacing points (the fifth line of one weft weave repeat unit) in the adjacent weft weave repeat unit. Therefore, compared with the satin weave, only half of PA conductive yarn warp interlacing points in one weft weave repeat unit can contact and separate with PET yarn in the adjacent weft weave repeat unit, which results in a smaller contact area between PA and PET. Therefore, the SWF-TENG with satin structure has higher output performances than that with twill weave. Different from a plain weave structure, no gap exists between two adjacent weft weave repeat units, which contributes to the least effective contact area and the lowest output performance.

To investigate the influences of structure parameters on the electrical output performances, including the PA conductive yarn fineness and the weft yarn density, several SWF-TENGs with satin weave and twill weave are fabricated and tested. Besides the PA conductive yarn which is 280D, PA conductive yarn which is 140D is also chosen as the electrode to fabricate the SWF-TENG. The electrical output performance comparisons between different SWF-TENGs fabricated with 140D and 280D PA conductive yarns are presented in Figure 7b,c, respectively. The electrical output measurements with 280D PA conductive yarn are significantly higher than that with 140D PA conductive yarn. Due to the decrease in conductive yarn fineness, the effective contact area between PA conductive yarn and PET decreases, and the outputs are lower.

Due to the creative fabricating method, the weft yarn density during the weaving process is relatively lower than the common weave structures. As a result of the lower weft yarn density and the big PU tensile strain (300%), the SWF-TENG is qualified with excellent stretchability. Therefore, the weft yarn density may affect the outputs of SWF-TENG. To further reveal the relationship between the weft yarn density and the electrical output performances, the SWF-TENGs with different weft yarn densities, including 250 ends/10 cm and 190 ends/10 cm, are fabricated and tested. As illustrated in Figure 7d, the electrical output performances of SWF-TENG with a bigger weft yarn density are higher than that with a smaller weft yarn density, due to the larger distance between the PA conductive yarn and the PET during stretching–releasing motion. The larger the weft yarn density, the bigger the distance between the PA conductive yarn and the PET, and the higher the electrical output performances.

To demonstrate the ability of SWF-TENG to harvest energy during press mode, the electrical output performances of the SWF-TENG with satin weave are measured by using a linear motor to provide periodic contact-separation movements. The V_OC_, I_SC_, and Q_SC_ are presented in Figure 8. When the contacted area is 4 cm^2^ and the press frequency increases from 0.5 Hz to 2.5 Hz, the V_OC_ and Q_SC_ remain almost the same, and the I_SC_ increases significantly from 12.9 nA to 26.1 nA with increasing the frequency. As shown in Figure 8b, the working mechanism of this SWF-TENG fabric is presented. In the original stage, no electrical potential exists between the surface of the PA conductive yarn and the PTFE film. By pressing PTFE film onto the fabric the surfaces of the PA conductive yarn and PTFE film are charged with the same amount of opposite charges (Figure 8(bi)). The PA conductive yarn proves to be positively charged because of the PTFE’s ability to attract more electrons than silver. When they separate from each other, a potential difference between the electrode and the ground prompts electrons flowing flow from the ground through external loading to the Ag (Figure 8(bii)), resulting in an electrical current. As the PTFE film is moving quite far away a new electrical equilibrium achieves, and the electrons stop moving (Figure 8(biii)). As the PTFE film approaches the PA conductive yarn again, electrons flow inversely from the electrode (PA conductive yarn) to the ground to achieve a charge balance (Figure 8(biv)). When the PTFE film fully contacts the PA conductive yarn, charge neutralization occurs again. Continuous contact-separation movements between the PTFE film and the PA conductive yarn generate continuous alternating current outputs from this SWF-TENG fabric through the external loading. To obtain a more quantitative understanding of the electricity-generating process, we establish a theoretical model to observe the electric potential distribution of PTFE film and SWF-TENG during the contact-separation movements by a simple finite element simulation using COMSOL Multiphysics (Figure 8c). In vertical contact-separate mode the surface charge density is set as 1.8 μC/m^2^. The vertical displacement is 2 mm. By weaving this flexible and stretchable fabric TENG into clothing, carpet, or curtain, we can easily harvest clean and sustainable mechanical energy to power lights or intelligent devices. As shown in Figure 8d, the SWF-TENG can be used to light up 34 LEDs marked as alphabets “KTC” by only hand tapping the fabric. As indicated above, the SWF-TENG has a high sensitivity to the external tensile strain. Due to the high detection precision of the SWF-TENG, it is chosen to be used as a bend–stretch sensor which is applied on the knee location of the human trouser to detect the gait, as shown in Figure 8e. When the tester is strolling, striding, and running, the output voltages of SWF-TENG as a bend–stretch sensor are recorded, as presented in Figure 8f. According to the result, all the curves present stable electrical signals due to the periodic bend–stretch movements of the human leg. When the tester is strolling, the SWF-TENG on the knee bends and stretches more than that during walking, resulting in a higher voltage output of strolling. Furthermore, the motion frequency of running is faster than that of strolling. The step number and motion speed can also be obtained according to the collected voltage signals of the three SWF-TENGs.

## 4. Conclusions

In summary, a stretchable woven fabric-based triboelectric nanogenerator (SWF-TENG) is designed. It is made of PA conductive yarn, PU, and PET multifilament. Based on the mature weaving technology, the stretchable SWF-TENG can be mass-manufactured on a high-speed and economical weaving frame, which decreases the production costs significantly and can promote the wide application of fabric-based TENG in wearable electronics. Due to the special woven technique, the SWF-TENG achieves 300% tensile strain by applying non-elastic yarns and elastic yarns (only one-fifth of total warp yarns), which improve the fabric comfortability and can be used as a self-powered stretchable sensor. The unique woven structure endows the SWF-TENG with a high contacting-separating area between yarns during stretching–releasing motion that does not need to rely on external substrates or surfaces to acquire sliding or contact-separating triboelectricity. Several SWF-TENGs with different structures, including plain, twill, and satin weave, are fabricated and tested to systematically investigate the influence of structure, material, tensile strain, and weft yarn density on the electrical output performances. As an energy harvester, the SWF-TENG is capable of lighting up 34 LEDs marked as alphabet “KTC” by only hand tapping the fabric. As a bend–stretch sensor, the SWF-TENG can detect and identify human gait. The SWF-TENGs not only have stable electrical output performances, high elasticity, excellent mechanical stability, and low cost, but can also respond rapidly to external mechanical stimuli, which can be used for self-powered sensing and energy harvesting. This mass-manufactured SWF-TENG provides a promising research orientation for clean power sources and self-powered stretchable sensors of textile-based TENG.

## Figures and Tables

**Figure 1 nanomaterials-13-00863-f001:**
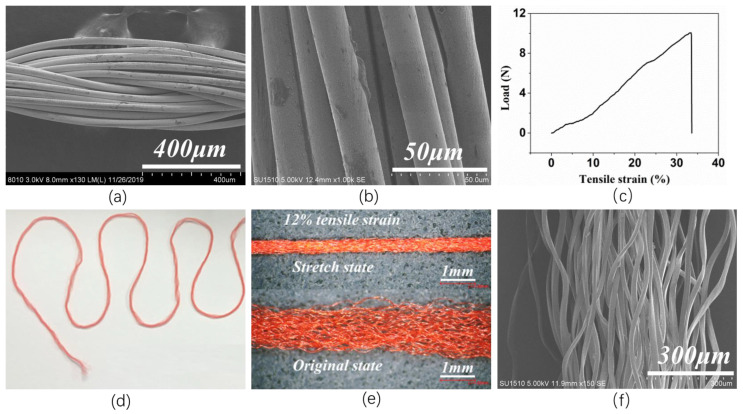
(**a**) Scanning electron microscopy (SEM) photograph of the PA conductive yarn; (**b**) Scanning electron microscopy (SEM) photograph of the Ag coating on the surface of the PA conductive yarn; PET multifilament; (**c**) The tensile property of the PA conductive yarn; (**d**) The digital photograph of the PET multifilament; (**e**) Optical microscopy (OM) photograph of the PET multifilament; (**f**) Scanning electron microscopy (SEM) photograph of the PET multifilament.

**Figure 2 nanomaterials-13-00863-f002:**
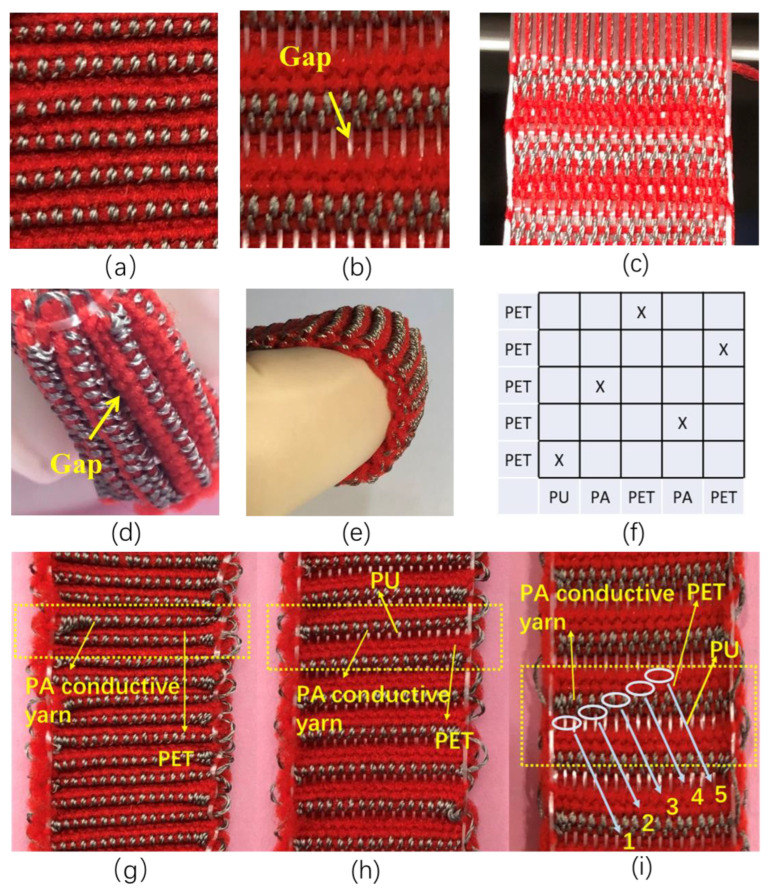
(**a**) SWF-TENG with satin weave before stretch; (**b**,**c**) SWF-TENG with satin weave after stretch; (**d**,**e**) The side view of SWF-TENG with satin weave; (**f**) Satin weave diagram; (**g**) SWF-TENG with satin weave before stretch; (**h**,**i**) SWF-TENG with satin weave after stretch.

**Figure 3 nanomaterials-13-00863-f003:**
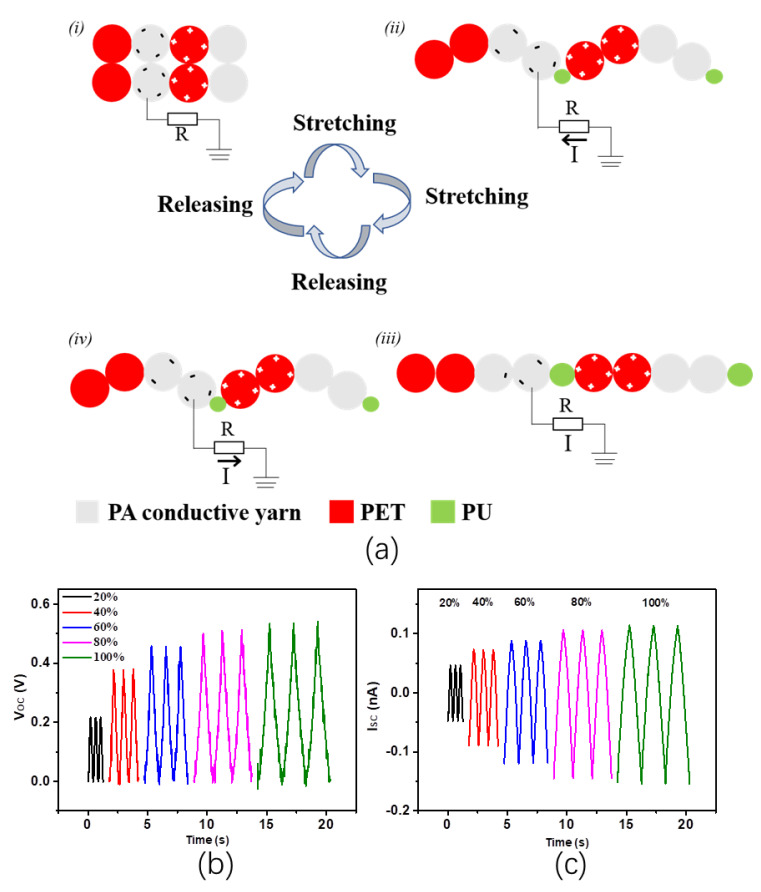
(**a**) The working mechanism of SWF-TENG with a satin weave, (**i**) In the original state without stretch; (**ii**) The SWF-TENG is gradually stretched; (**iii**) The SWF-TENG is stretched to the largest deformation; (**iv**) The SWF-TENG is recovering to the original state; (**b**,**c**) Electrical output performances of SWF-TENG with satin structure.

**Figure 4 nanomaterials-13-00863-f004:**
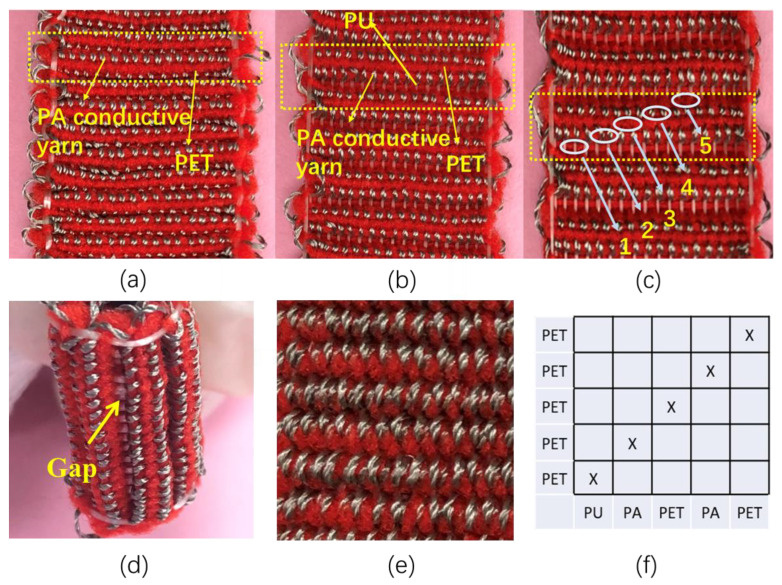
(**a**) The digital photograph of SWF-TENG with twill weave before stretch; (**b**,**c**) SWF-TENG with twill weave after stretch; (**d**) The side view of SWF-TENG with twill weave; (**e**) The back side of SWF-TENG with twill weave; (**f**) The weave diagram.

**Figure 5 nanomaterials-13-00863-f005:**
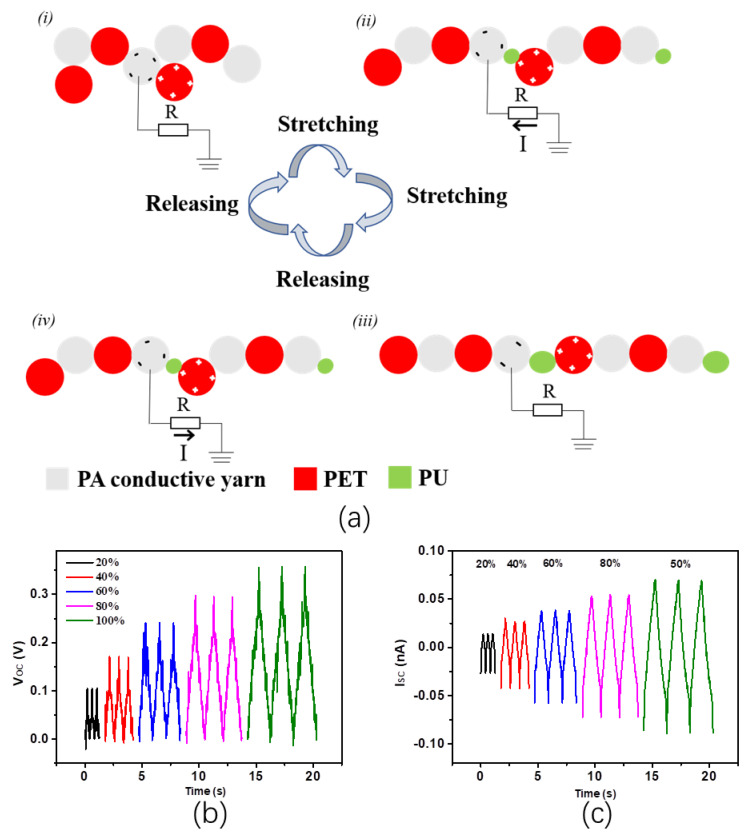
(**a**) The working mechanism of SWF-TENG with twill weave, (**i**) In the original state without stretch; (**ii**) The SWF-TENG is gradually stretched; (**iii**) The SWF-TENG is stretched to the largest deformation; (**iv**) The SWF-TENG is recovering to the original state; (**b**,**c**) Electrical output performances of SWF-TENG with twill structure.

**Figure 6 nanomaterials-13-00863-f006:**
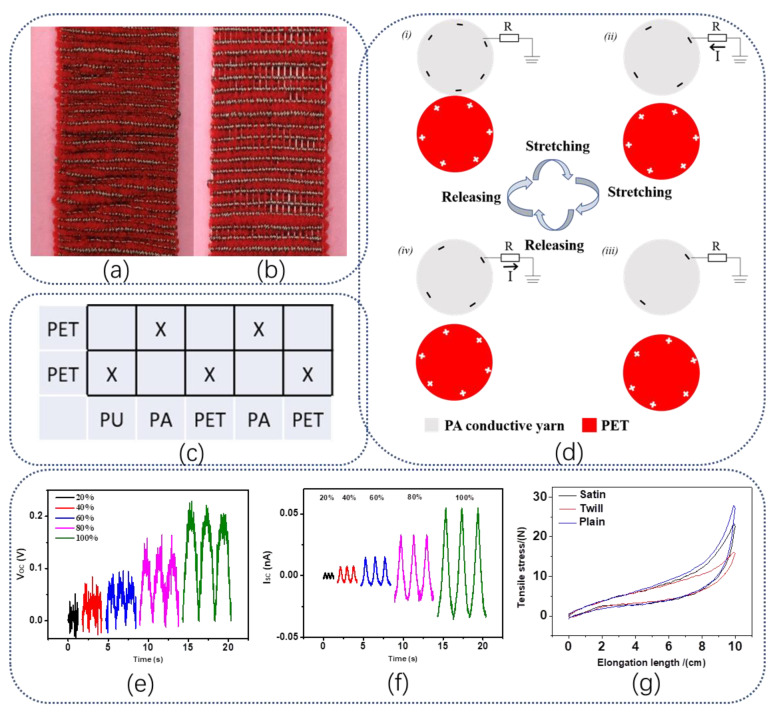
(**a**,**b**) Digital photographs of SWF-TENG with plain weave before and after stretch; (**c**) The weave diagram of plain structure; (**d**) The working mechanism of SWF-TENG with plain structure, (**i**) The SWF-TENG is stretched at the largest deformation state; (**ii**) The SWF-TENG is gradually recovering; (**iii**) The SWF-TENG recovers to the original state; (**iv**) The SWF-TENG is gradually stretched. (**e**,**f**) Electrical output performances of SWF-TENG with plain structure; (**g**) The tensile property of the SWF-TENG with different structures.

**Figure 7 nanomaterials-13-00863-f007:**
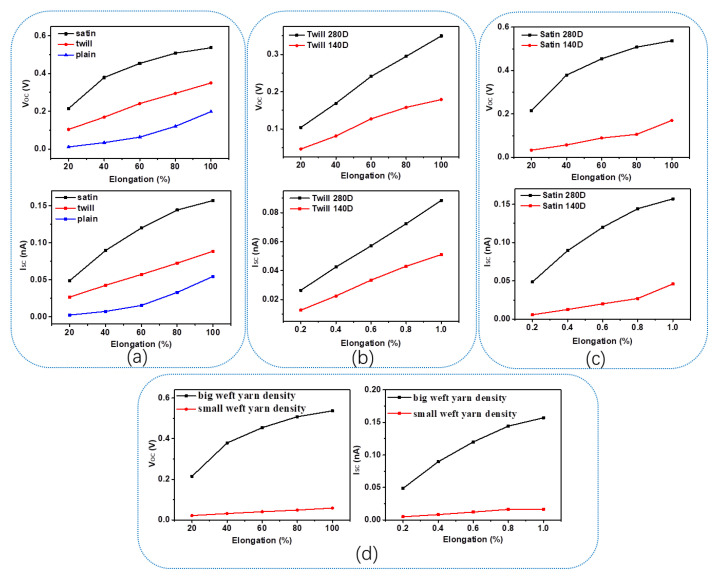
(**a**) The outputs comparison of satin, twill, and plain structures; The electrical output performances comparison between different SWF-TENGs; (**b**) Twill structure fabricated by the PA conductive yarn, including 280D and 140D; (**c**) Satin structure fabricated by the PA conductive yarn, including 280D and 140D; (**d**) SWF-TENGs with different weft yarn densities.

**Figure 8 nanomaterials-13-00863-f008:**
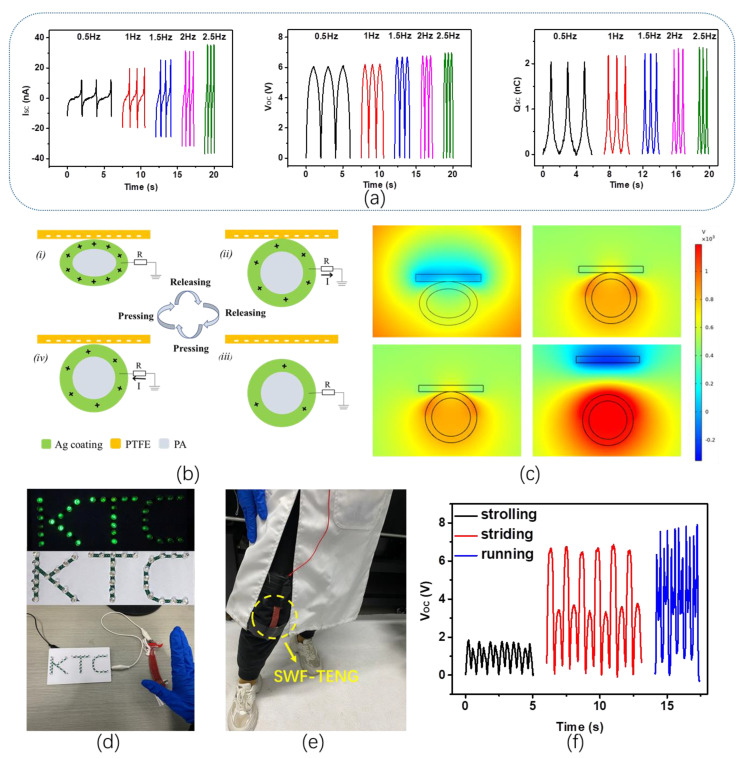
(**a**) The electrical output performances of SWF-TENG under press mode with different frequencies; (**b**) The working mechanism of SWF-TENG under press mode, (**i**) In the original state with pressure between the PTFE film and the fabric; (**ii**) The SWF-TENG and the PTFE film are gradually separating; (**iii**) The PTFE film is moving quite far away from the SWF-TENG; (**iv**) The PTFE film is contacting the SWF-TENG gradually; (**c**) Simulated electric field distribution of SWF-TENG under press mode; (**d**) Demonstration of lighting up 34 LEDs marked as alphabets “KTC” by only hand tapping the fabric TENG; (**e**) The testing interface of the SWY-TENG as a real-time bend–stretch sensor; (**f**) V_OC_ of the SWY-TENG as a real-time bend–stretch sensor.

## Data Availability

Not applicable.

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
