# Peer review of "Stretchable Woven Fabric-Based Triboelectric Nanogenerator for Energy Harvesting and Self-Powered Sensing"

_nanomaterials, 2023, doi:10.3390/nano13050863_

Round 1
Reviewer 1 Report
This work, titled “Stretchable woven fabric-based triboelectric nanogenerator for energy harvesting and self-powered sensing”, describes the fabrication and characterization of a highly stretchable woven fabric-based triboelectric nanogenerator. It is fabricated using a multifilament fabrication technique by using a succession of PA conductive yarn, PU and PET materials with different combinations and exploiting the intrinsic triboelectric features of the final achieved device. Different structures have been fabricated and tested, including plain, twill, and satin weave, and a systematic investigation has been carried out to analyse the influence of structure, material, tensile strain, and weft yarn density on the electrical output performances. The authors demonstrated significant results by exhaustive experimentation, showing these fabrics not only have stable electrical output performances, high elasticity, excellent mechanical stability, and low cost but also good responsivity to rapidly changing external mechanical stimuli, allowing them to be used for a self-powered sensing and energy harvesting system together.
The reviewer found this work interesting and useful. However, major revisions are needed before being further considered for publication in Nanomaterials. For these reasons, the reviewer suggests the authors address the following comments:
1. In lines 68 -69 of page 2, the authors mention some materials such as PU, PA and PET. Since they appear for the first time in the text, the reviewer suggests expanding the acronym before mentioning them.
2. In figure 5, the output performances of the fabricated devices have been shown. Nevertheless, the plots shown in the figure are not consistent with the text. In lines 225 – 227, the authors stated that “According to the comparison of these three SWF-TENGs’ outputs which is shown in Figure 5h, it’s obvious that SWF-TENG with satin weave has the highest electrical output performances, and SWF-TENG with plain weave has the lowest electrical output performances.” even if from the figure, according to the caption, the plain structure seems to have the best performances. Please clarify this point.
3. Moreover, the figures showing the working mechanisms of the TENG can be improved. The reviewer suggests separating the output response of each structure and putting the output plot (current and voltage) in the corresponding figure for a better description. An additional figure for the final comparison could help clarify the performances of the fabricated devices.
4. In figure 5, the authors described the performances of the TENGs showing the output response as a function of the elongation (%). How did they achieve the output response of the device as a function of the elongation? Has a conversion from tensile strain to elongation been carried out?
5. To test the stretchability of the fabric-based TENG, have measurements of the elastic modulus of the final devices (all the structures) been carried out?
6. In line 293, the authors stated the COMSOL software has been used to perform FEM simulations. In the text, the sentence “a simple finite element simulation using COMSOL Multiphasic” needs to be corrected saying “a simple finite element simulation using COMSOL Multiphysics”. Moreover, additional information and details should be given about the FEM simulation (3D model, mesh and element type, physics, etc…).
Moreover, please address the following minor revisions:
1. In Figure 1, please make the font size and style larger;
2. In Figures 2, 3, 4 and 5, the pictures showing the real device are often out of focus. Could the authors please put more clear, not-blurred pictures? Moreover, please make font size and style larger;
3. In lines 61 – 63, the sentence “In this work, several stretchable woven fabric-based TENGs (SWF-TENG) are designed and presented that is capable of harvesting mechanical energy and tensile strain sensing.”, the verb “to be” should follow the plural form of the subject.
4. In lines 127 – 128, the definite article “The” must be changed with “the” (lowercase letter);
Reviewer 2 Report
The article entitled “Stretchable woven fabric-based triboelectric nanogenerator for energy harvesting and self-powered sensing” by Chen et. al., is well presented article on an interesting topic of fabric-based triboelectric nanogenerator. It may be accepted for publication after few minor revisions.
(1) In the “introduction” section, it is better to provide full forms for PU, PA and PET.
(2) There is a discrepancy between the figure caption (1b) and corresponding text.
Reviewer 3 Report
The paper entitled “Stretchable woven fabric-based triboelectric nanogenerator for energy harvesting and self-powered sensing,” authored by Chen et al., developed a stretchable woven fabric platform as a dual-purpose nanogenerator for mountable sensors and energy harvesting. Even though the topic of the paper is of interest, however, it requires serious modifications prior to possible acceptance or publication in the journal. My comments for the paper are as follows:
1. The paper’s writing style requires improvement, and the authors should enhance the storyline and follow-up of the paper.
2. In the abstract, please provide information about the devised nanogenerator and its components and refer to noteworthy outcomes.
3. In the introduction section, please refer to previous works in the field and compare the outcome of your devised nanogenerator and/or sensor with previous practices.
4. The materials and methods are missing too much information about used materials, fabrics, methods of preparation and characterization techniques, and their details. Additionally, authors must not bring any characterization in materials and methods, and all of them must be moved to the results and discussion.
5. From an expert point of view, the paper lacks in-depth explanation and information regarding the sensor's sensitivity, the efficiency of the nanogenerator, and the tensile test showing the reversible stretchability must be provided within the text.
6. The preparation method of the SWF-TENG must be provided in the materials and methods with full details.
7. More scientific data are required to showcase the nanogenerator and sensor performance.
8. Please improve Figure 5 and its features.
9. The conclusion can not cover the noteworthy outcomes of the paper; hence, please improve it accordingly.
Round 2
Reviewer 1 Report
No additional comments for authors. The reviewer suggests to accept the manuscript in the present form.